# Regularized Anderson Acceleration for Off-Policy Deep Reinforcement Learning

**Wenjie Shi, Shiji Song, Hui Wu, Ya-Chu Hsu, Cheng Wu, Gao Huang**[*]

Department of Automation, Tsinghua University, Beijing, China

Beijing National Research Center for Information Science and Technology (BNRist)

`{shiwj16, wuhui14, xuyz17}@mails.tsinghua.edu.cn`
`{shijis, wuc, gaohuang}@tsinghua.edu.cn`

## Abstract

Model-free deep reinforcement learning (RL) algorithms have been widely used for a range of complex control tasks. However, slow convergence and sample inefficiency remain challenging problems in RL, especially when handling continuous and high-dimensional state spaces. To tackle this problem, we propose a general acceleration method for model-free, off-policy deep RL algorithms by drawing the idea underlying regularized Anderson acceleration (RAA), which is an effective approach to accelerating the solving of fixed point problems with perturbations. Specifically, we first explain how policy iteration can be applied directly with Anderson acceleration. Then we extend RAA to the case of deep RL by introducing a regularization term to control the impact of perturbation induced by function approximation errors. We further propose two strategies, i.e., progressive update and adaptive restart, to enhance the performance. The effectiveness of our method is evaluated on a variety of benchmark tasks, including Atari 2600 and MuJoCo. Experimental results show that our approach substantially improves both the learning speed and final performance of state-of-the-art deep RL algorithms. The code and models are available at: https://github.com/shiwj16/raa-drl.

## 1 Introduction

Reinforcement learning (RL) is a principled mathematical framework for experience-based autonomous learning of policies. In recent years, model-free deep RL algorithms have been applied in a variety of challenging domains, from game playing [1, 2] to robot navigation [3, 4]. However, sample inefficiency, i.e., the required number of interactions with the environment is impractically high, remains a major limitation of current RL algorithms for problems with continuous and high-dimensional state spaces. For example, many RL approaches on tasks with low-dimensional state spaces and fairly benign dynamics may even require thousands of trials to learn. Sample inefficiency makes learning in real physical systems impractical and severely prohibits the applicability of RL approaches in more challenging scenarios.

A promising way to improve the sample efficiency of RL is to learn models of the underlying system dynamics. However, learning models of the underlying transition dynamics is difficult and inevitably leads to modelling errors. Alternatively, off-policy algorithms such as deep Q-learning (DQN) [1] and its variants [5, 6], deep deterministic policy gradient (DDPG) [7], soft actor-critic (SAC) [8, 9] and off-policy hierarchical RL [10], which instead aim to reuse past experience, are commonly used to alleviate the sample inefficiency problem. Unfortunately, off-policy algorithms are typically based on policy iteration or value iteration, which repeatedly apply the Bellman operator of interest and generally require an infinite number of iterations to converge exactly to the optima. Moreover, the Bellman iteration constructs a contraction mapping which converges asymptotically to the optimal

---

[*]Corresponding auther.

value function [11]. Iterating this mapping essentially results in a fixed-point problem [12] and thus may be unacceptably slow to converge. These issues are further exacerbated when nonlinear function approximator such as neural network is utilized or the tasks have continuous state and action spaces.

This paper explores how to accelerate the convergence or improve the sample efficiency for model-free, off-policy deep RL. We make the observation that RL is closely linked to fixed-point iteration: the optimal policy can be found by solving a fixed-point problem of associated Bellman operator. Therefore, we attempt to embrace the idea underlying Anderson acceleration (also known as Anderson mixing, Pulay mixing) [13, 14], which is a method capable of speeding up the computation of fixed-point iterations. While the classic fixed-point iteration repeatedly applies the operator to the last estimate, Anderson acceleration searches for the optimal point that has minimal residual within the subspace spanned by several previous estimates, and then applies the operator to this optimal estimate. Prior work [15] has successfully applied Anderson acceleration to value iteration and preliminary experiments show a significant speed up of convergence. However, existing application is only feasible on simple tasks with low-dimensional, discrete state and action spaces. Besides, as far as we know, Anderson acceleration has never been applied to deep RL due to some long-standing issues including biases induced by sampling a minibatch and function approximation errors.

In this paper, Anderson acceleration is first applied to policy iteration under a tabular setting. Then, we propose a practical acceleration method for model-free, off-policy deep RL algorithms based on regularized Anderson acceleration (RAA) [16], which is a general paradigm with a Tikhonov regularization term to control the impact of perturbations. The structure of perturbations could be the noise injected from the outside and high-order error terms induced by a nonlinear fixed-point iteration function. In the context of deep RL, function approximation errors are major perturbation source for RAA. We present two bounds to characterize how the regularization term controls the impact of function approximation errors. Two strategies, i.e., progressive update and adaptive restart, are further proposed to enhance the performance. Moreover, our acceleration method can be implemented readily to deep RL algorithms including Dueling-DQN [5] and twin delayed DDPG (TD3) [17] to solve very complex, high-dimensional tasks, such as Atari 2600 and MuJoCo [18] benchmarks. Finally, the empirical results show that our approach exhibits a substantial improvement in both learning speed and final performance over vanilla deep RL algorithms.

## 2   Related Work

Prior works have made a number of efforts to improve the sample efficiency and speed up the convergence of deep RL from different respects, such as variance reduction [19, 20], model-based RL [21, 22, 23], guided exploration [24, 25], etc. One of the most widely used techniques is off-policy RL, which combines temporal difference [26] and experience replay [27, 28] so as to make use of all the previous samples before each update to the policy parameters. Though introducing biases by using previous samples, off-policy RL alleviates the high variance in estimation of Q-value and policy gradient [29]. Consequently, fast convergence is rendered when under fine parameter-tuning.

As one kernel technique of off-policy RL, temporal difference is derived from the Bellman iteration which can be regarded as a fixed-point problem [12]. Our work focuses on speeding up the convergence of off-policy RL via speeding up the convergence of the eesential fixed-point problem, and replying on a technique namely Anderson acceleration. This method is exploited by prior work [13, 30] to accelerate the fixed-point iteration by computing the new iteration as linear combination of previous evaluations. In the linear case, the convergence rate of Anderson acceleration has been elaborately analyzed and proved to be equal to or better than fixed-point iteration in [14]. For nonlinear fixed-point iteration, regularized Anderson acceleration is proposed by [16] to constrain the norm of coefficient vector and reduce the impact of perturbations. Recent works [15, 31] have applied the Anderson acceleration to value iteration and deep neural network, and preliminary experiments show that a significant speedup of convergence is achieved. However, there is still no research showing its acceleration effect on deep RL for complex high-dimensional problems, as far as we know.

## 3   Preliminaries

Under RL paradigm, the interaction between an agent and the environment is described as a Markov Decision Process (MDP). Specifically, at a discrete timestamp $t$, the agent takes an action $a_t$ in a state $s_t$ and transits to a subsequent state $s_{t+1}$ while obtaining a reward $r_t = r(s_t, a_t)$ from the environment. The transition between states satisfies the Markov property, i.e., $P(s_{t+1}|s_t, a_t, \ldots, s_0, a_0) = P(s_{s+t}|s_t, a_t)$. Usually, the RL algorithm aims to search a policy $\pi(a|s)$ that maximizes the expected

sum of discounted future rewards. Q-value function describes the expected return starting from a state-action pair $(s, a)$: $Q^\pi(s, a) = \mathbb{E}\left[\sum_{t=0}^{\infty} \gamma^t r_{t+1} | s_0 = s, a_0 = a\right]$, where the policy $\pi(a|s)$ is a function or conditional distribution mapping the state space $\mathcal{S}$ to the action space $\mathcal{A}$.

## 3.1 Off-policy reinforcement learning

Most off-policy RL algorithms are derived from policy iteration, which alternates between policy evaluation and policy improvement to monotonically improve the policy and the value function until convergence. For complex environments with unknown dynamics and continuous spaces, policy iteration is generally combined with function approximation, and parameterized Q-value function (or critic) and policy function are learned from sampled interactions with environment. Since critic is represented as parameterized function instead of look-up table, the policy evaluation is replaced with an optimization problem which minimizes the squared temporal difference error, the discrepancy between the outputs of critics after and before applying the Bellman operator

$$L(\theta) = \mathbb{E}\left[((\mathcal{T}Q_{\theta'})(s, a) - Q_\theta(s, a))^2\right], \tag{1}$$

where typically the Bellman operator is applied to a separate target value network $Q_{\theta'}$ whose parameter is periodically replaced or softly updated with copy of current Q-network weight.

In off-policy RL field, prior works have proposed a number of modifications on the Bellman operator to alleviate the overestimation or function approximation error problems and thus achieved significant improvement. Similar to policy improvement, DQN replaces the current policy with a greedy policy for the next state in the Bellman operator

$$(\mathcal{T}Q_{\theta'})(s_t, a_t) = \mathbb{E}_{s_{t+1}, r_t}\left[r(s_t, a_t) + \gamma \max_a Q_{\theta'}(s_{t+1}, a)\right]. \tag{2}$$

As the state-of-the-art actor-critic algorithm for continuous control, TD3 [17] proposes a clipped double Q-learning variant and a target policy smoothing regularization to modify the Bellman operator, which alleviates overestimation and overfitting problems,

$$(\mathcal{T}Q_{\theta'})(s_t, a_t) = \mathbb{E}_{s_{t+1}, r_t}\left[r(s_t, a_t) + \gamma \min_{j=1,2} Q_{\theta'_j}(s_{t+1}, \pi_{\phi'}(s_{t+1}) + \epsilon)\right], \tag{3}$$

where $Q_{\theta_j}(s, a)(j = 1, 2)$ denote two critics with decoupled parameters $\theta_j$. The added noise $\epsilon \sim \mathrm{clip}(\mathcal{N}(0, \sigma), -c, c)$ is clipped by the positive constant $c$.

## 3.2 Anderson acceleration for value iteration

Most RL algorithms are derived from a fundamental framework named policy iteration which consists of two phases, i.e. policy evaluation and policy improvement. The policy evaluation estimates the Q-value function induced by current policy by iterating a Bellman operator from an initial estimate. Following the policy evaluation, the policy improvement acquires a better policy from a greedy strategy, The policy iteration alternates two phases to update the Q-value and the policy respectively until convergence. As a special variant of policy iteration, value iteration merges policy evaluation and policy improvement into one iteration

$$V_{k+1}(s) \leftarrow (\mathcal{T}V_k)(s) = \max_a \mathbb{E}_{s', r}\left[r + \gamma V_k(s')\right], \forall s \in \mathcal{S}, \tag{4}$$

and iterates it until convergence from a initial $V_0$, where the Bellman operation is only repeatedly applied to the last estimate. Anderson acceleration is a widely used technique to speed up the convergence of fixed-point iterations and has been successfully applied to speed up value iteration [15] by linearly combining previous $m \ (m > 1)$ value estimates,

$$V_{k+1} \leftarrow \sum_{i=1}^{m} \alpha_i^k \mathcal{T}V_{k-m+i}, \tag{5}$$

where the coefficient vector $\alpha^k \in \mathbb{R}^m$ is determined by minimizing the norm of total Bellman residuals of these estimates,

$$\alpha^k = \underset{\alpha \in \mathbb{R}^m}{\arg\min} \left\|\sum_{i=1}^{m} \alpha_i(\mathcal{T}V_{k-m+i} - V_{k-m+i})\right\|, \quad \text{s.t.} \sum_{i=1}^{m} \alpha_i = 1. \tag{6}$$

For the $\ell_2$-norm, the minimum can be analytically solved by using the Karush-Kuhn-Tucker conditions. Corresponding coefficient vector is given by

$$\alpha^k = \frac{(\Delta_k^T \Delta_k)^{-1}\mathbf{1}}{\mathbf{1}^T(\Delta_k^T \Delta_k)^{-1}\mathbf{1}}, \tag{7}$$

where $\Delta_k = [\delta_{k-m+1}, \dots \delta_k] \in \mathbb{R}^{|\mathcal{S}| \times m}$ is a Bellman residuals matrix with $\delta_i = \mathcal{T}V_i - V_i \in \mathbb{R}^{|\mathcal{S}|}$, and $\mathbf{1} \in \mathbb{R}^m$ denotes the vector with all components equal to one [15].

# 4 Regularized Anderson Acceleration for Deep Reinforcement Learning

Our regularized Anderson acceleration (RAA) method for deep RL can be derived starting from a direct implementation of Anderson acceleration to the classic policy iteration algorithm. We will first present this derivation to show that the resulting algorithm converges faster to the optimal policy than the vanilla form. Then, a regularized variant is proposed for a more general case with function approximation. Based on this theory, a progressive and practical acceleration method with adaptive restart is presented for off-policy deep RL algorithms.

## 4.1 Anderson acceleration for policy iteration

As described above, Anderson acceleration can be directly applied to value iteration. However, policy iteration is more fundamental and suitable to scale to deep RL, compared to value iteration. Unfortunately, the implementation of Anderson acceleration is complicated when considering policy iteration, because there is no explicit fixed-point mapping between the policies in any two consecutive steps, which make it impossible to straightforwardly apply Anderson acceleration to the policy $\pi$.

Due to the one-to-one mapping between policies and Q-value functions, policy iteration can be accelerated by applying Anderson acceleration to the policy improvement, which establishes a mapping from the current Q-value estimate to the next policy. In this section, our derivation is based on a tabular setting, to enable theoretical analysis. Specifically, for the prototype policy iteration, suppose that estimates have been computed up to iteration $k$, and that in addition to the current estimate $Q^{\pi_k}$, the $m-1$ previous estimates $Q^{\pi_{k-1}}, ..., Q^{\pi_{k-m+1}}$ are also known. Then, a linear combination of estimates $Q^{\pi_i}$ with coefficients $\alpha_i$ [2] reads

$$Q_\alpha^k = \sum_{i=1}^m \alpha_i Q^{\pi_{k-m+i}} \text{ with } \sum_{i=1}^m \alpha_i = 1. \tag{8}$$

Due to this equality constraint, we define *combined Bellman operator* $\mathcal{T}_c$ as follows

$$\mathcal{T}_c Q_\alpha^k = \sum_{i=1}^m \alpha_i \mathcal{T} Q^{\pi_{k-m+i}}. \tag{9}$$

Then, one searches a coefficient vector $\alpha^k$ that minimizes the following objective function $J$ defined as the combined Bellman residuals among the entire state-action space $\mathcal{S} \times \mathcal{A}$,

$$\alpha^k = \underset{\alpha \in \mathbb{R}^m}{\operatorname{argmin}} J(\alpha) = \underset{\alpha \in \mathbb{R}^m}{\operatorname{argmin}} \left\| \sum_{i=1}^m \alpha_i (\mathcal{T} Q^{\pi_{k-m+i}} - Q^{\pi_{k-m+i}}) \right\|, \text{ s.t. } \sum_{i=1}^m \alpha_i = 1. \tag{10}$$

In this paper, we will consider the $\ell_2$-norm, although a different norm may also be feasible (for example $\ell_1$ and $\ell_\infty$, in which case the optimization problem becomes a linear program). The solution to this optimization problem is identical to (7) except that $\Delta_k = [\delta_{k-m+1}, ..., \delta_k] \in \mathbb{R}^{|\mathcal{S} \times \mathcal{A}| \times m}$ with $\delta_i = \mathcal{T} Q^{\pi_i} - Q^{\pi_i} \in \mathbb{R}^{|\mathcal{S} \times \mathcal{A}|}$. Detailed derivation can be found in Appendix A.1 of the supplementary material. Then, the new policy improvement steps are given by

$$\pi_{k+1}(s) = \underset{a}{\operatorname{argmax}} Q_\alpha^k(s, a) = \underset{a}{\operatorname{argmax}} \sum_{i=1}^m \alpha_i^k Q^{\pi_{k-m+1}}(s, a), \forall s \in \mathcal{S}. \tag{11}$$

Meanwhile, Q-value estimate $Q^{\pi_{k+1}}$ can be obtained by iteratively applying the following policy evaluation operator by starting from some initial function $Q_0$,

$$Q_i(s, a) \leftarrow \mathbb{E}_{s\prime, r} \left[ r + \gamma \mathbb{E}_{a\prime \sim \pi_{k+1}} [Q_{i-1}(s', a')] \right], \forall (s, a) \in (\mathcal{S}, \mathcal{A}). \tag{12}$$

In fact, the effect of acceleration can be explained intuitively. The linear combination $Q_\alpha^k$ is a better estimate of Q-value than the last one $Q^{\pi_k}$ in terms of combined Bellman residuals. Accordingly, the policy is improved from a better policy baseline corresponding to the better estimate of Q-value.

## 4.2 Regularized variant with function approximation

For RL control tasks with continuous state and action spaces, or high-dimensional state space, we generally consider the case in which Q-value function is approximated by a parameterized function approximator. If the approximation is sufficiently good, it might be appropriate to use it in place of $Q^\pi$ in (8)-(12). However, there are several key challenges when implementing Anderson acceleration with function approximation.

First, notice that the Bellman residuals in (10) are calculated among the entire state-action space. Unfortunately, sweeping entire state-action space is intractable for continuous RL, and a fine grained discretization will lead to the curse of dimensionality. A feasible alternative to avoid this issue is to use a sampled Bellman residuals matrix $\widetilde{\Delta}_k$ instead. To alleviate the bias induced by sampling a minibatch, we adopt a large sample size $N_A$ specifically for Anderson acceleration.

Second, function approximation errors are unavoidable and lead to biased solution of Anderson acceleration. The intricacies of this issue will be exacerbated by deep models. Therefore, function approximation errors will induce severe perturbation when implementing Anderson acceleration to policy iteration with function approximation. In addition to the perturbation, the solution (7) contains the inverse of a squared Bellman residuals matrix, which may suffer from ill-conditioning when the squared Bellman residuals matrix is rank-deficient, and this is a major source of numerical instability in vanilla Anderson acceleration. In other words, even if the perturbation is small, its impact on the solution can be arbitrarily large.

Under the above observations, we scale the idea underlying RAA to the policy iteration with function approximation in this section. Then, the coefficient vector (10) is now adjusted to $\widetilde{\alpha}^k$ that minimizes the perturbed objective function added with a Tikhonov regularization term,

$$\widetilde{\alpha}^k = \underset{\alpha \in \mathbb{R}^m}{\mathrm{argmin}} \left\| \sum_{i=1}^m \alpha_i (\mathcal{T} Q^{\pi_{k-m+i}} - Q^{\pi_{k-m+i}} + e_{k-m+i}) \right\| + \lambda \|\alpha\|^2, \ \text{ s.t. } \sum_{i=1}^m \alpha_i = 1, \quad (13)$$

where $e_{k-m+i}$ represents the perturbation induced by function approximation errors. The solution to this regularized optimization problem can be obtained analytically similar to (10),

$$\widetilde{\alpha}^k = \frac{(\widetilde{\Delta}_k^T \widetilde{\Delta}_k + \lambda I)^{-1} \mathbf{1}}{\mathbf{1}^T (\widetilde{\Delta}_k^T \widetilde{\Delta}_k + \lambda I)^{-1} \mathbf{1}}, \quad (14)$$

where $\lambda$ is a positive scalar representing the scale of regularization. $\widetilde{\Delta}_k = [\widetilde{\delta}_{k-m+1}, ..., \widetilde{\delta}_k] \in \mathbb{R}^{N_A \times m}$ is the sampled Bellman residuals matrix with $\widetilde{\delta}_i = \mathcal{T} Q^{\pi_i} - Q^{\pi_i} + e_i \in \mathbb{R}^{N_A}$.

In fact, the regularization term controls the norm of coefficient vector produced by RAA and reduces the impact of perturbation induced by function approximation errors, as shown analytically by the following proposition.

**Proposition 1.** *Consider two identical policy iterations $\mathcal{I}_1$ and $\mathcal{I}_2$ with function approximation. $\mathcal{I}_2$ is implemented with regularized Anderson acceleration and takes into account approximation errors, whereas $\mathcal{I}_1$ is only implemented with vanilla Anderson acceleration. Let $\alpha^k$ and $\widetilde{\alpha}^k$ be the coefficient vectors of $\mathcal{I}_1$ and $\mathcal{I}_2$ respectively. Then, we have the following bounds*

$$\|\widetilde{\alpha}^k\| \leq \sqrt{\frac{\lambda + \|\widetilde{\Delta}_k\|^2}{m\lambda}}, \quad \|\widetilde{\alpha}^k - \alpha^k\| \leq \frac{\|\widetilde{\Delta}_k^T \widetilde{\Delta}_k - \Delta_k^T \Delta_k\| + \lambda}{\lambda} \|\alpha^k\|. \quad (15)$$

*Proof.* See Appendix A.2 of the supplementary material. $\square$

From the above bounds, we can observe that regularization allows a better control of the impact of function approximation errors, but also causes an inevitable gap between $\widetilde{\alpha}^k$ and $\alpha^k$. Qualitatively, large regularization scale $\lambda$ means less impact of function approximation errors. On the other hand, overlarge $\lambda$ leads to very small norm of coefficient vector $\widetilde{\alpha}^k$, which means the coefficients for previous estimates is nearly identical. However, according to (10), equal coefficients are probably far away from the optima $\alpha^k$ and thus result in great performance loss of Anderson acceleration.

### 4.3 Implementation on off-policy deep reinforcement learning

As discussed in last section, it is impossible to directly use policy iteration in very large continuous domains. To that end, most off-policy deep RL algorithms apply the mechanism underlying policy iteration to learn approximations to both the Q-value function and the policy. Instead of iterating policy evaluation and policy improvement to convergence, these off-policy algorithms alternate between optimizing two networks with stochastic gradient descent. For example, actor-critic method is a well-known implementation of this mechanism. In this section, we show that RAA for policy iteration can be readily extended to existing off-policy deep RL algorithms for both discrete and continuous control tasks, with only a few modifications to the update of critic.

**Algorithm 1:** RAA-Dueling-DQN Algorithm

---

Initialize a critic network $Q_\theta$ with random parameters $\theta$;
Initialize $m$ target networks $\theta^i \leftarrow \theta$ $(i = 1, ..., m)$ and replay buffer $\mathcal{D}$;
Initialize restart checking period $T_r$ and maximum training steps $K$;
Set $k = 0$, $c_1 = 1$, $\Delta_{min} = \inf$, $\Delta_{T_r} = 0$;
**while** $k < K$ **do**

> Receive initial observation state $s_0$;
> **for** $t = 1$ *to* $T$ **do**
>
> > Set $k = k + 1$, and $m_k = \min(c_k, m)$;
> > With probability $\varepsilon$ select a random action $a_t$, otherwise select $a_t = \operatorname{argmax}_a Q_\theta(s_t, a)$;
> > Execute $a_t$, receive $r_t$ and $s_{t+1}$, store transition $(s_t, a_t, r_t, s_{t+1})$ into $\mathcal{D}$;
> > Sample minibatch of transitions $(s, a, r, s')$ from $\mathcal{D}$;
> > Perform Anderson acceleration steps (13)-(14) and obtain $\widetilde{\alpha}^k$, $\Delta_{T_r} = \Delta_{T_r} + \|\widetilde{\delta}_k\|_2^2$;
> > Update the critic by minimizing the loss function (18) with $y_t$ equal to the RHS of (17);
> > Update target networks every $M$ steps: $\theta^i \leftarrow \theta^{i+1}$ $(i = 1, ..., m - 1)$ and $\theta^m \leftarrow \theta$;
> > $c_{k+1} = c_k + 1$;
> > **if** $k \mod T_r = 0$ **then**
> >
> > > $\Delta_{min} = \min(\Delta_{min}, \Delta_{T_r})$;
> > > **if** $\Delta_{T_r} > \Delta_{min}$ **then**
> > >
> > > > $\Delta_{min} = \inf$, and $c_{k+1} = 1$;

---

### 4.3.1 Regularized Anderson acceleration for actor-critic

Consider a parameterized Q-value function $Q_\theta(s_t, a_t)$ and a tractable policy $\pi_\phi(a_t|s_t)$, the parameters of these networks are $\theta$ and $\phi$. In the following, we first give the main results of RAA for actor-critic. Then, RAA is combined with Dueling-DQN and TD3 respectively.

Under the paradigm of off-policy deep RL (actor-critic), RAA variant of policy iteration (11)-(12) degrades into the following Bellman equation

$$Q_\theta(s_t, a_t) = \mathbb{E}_{s_{t+1}, r_t} \left[ r_t + \gamma \sum_{i=1}^{m} \widetilde{\alpha}_i \max_{a_{t+1}} Q_{\theta^i}(s_{t+1}, a_{t+1}) \right], \tag{16}$$

where $\theta^i$ is the parameters of target network before $i$ update steps. Furthermore, to mitigate the instability resulting from drastic update step of Anderson acceleration, the following progressive Bellman equation (or progressive update) with RAA is used practically,

$$Q_\theta(s_t, a_t) = \beta \sum_{i=1}^{m} \widetilde{\alpha}_i Q_{\theta^i}(s_t, a_t) + (1 - \beta) \mathbb{E}_{s_{t+1}, r_t} \left[ r_t + \gamma \sum_{i=1}^{m} \widetilde{\alpha}_i \max_{a_{t+1}} Q_{\theta^i}(s_{t+1}, a_{t+1}) \right], \tag{17}$$

where $\beta$ is a small positive coefficient.

Generally, the loss function of critic is then formulated as the following squared consistency error of Bellman equation,

$$L_Q(\theta) = \mathbb{E}_{(s_t, a_t) \in \mathcal{D}} \left[ (Q_\theta(s_t, a_t) - y_t)^2 \right], \tag{18}$$

where $\mathcal{D}$ is the distribution of previously sampled transitions, or a replay buffer. The target value of Q-value function or critic is represented by $y_t$.

**RAA-Dueling-DQN.** Different from vanilla Dueling-DQN algorithm using general Bellman equation, we instead use progressive Bellman equation with RAA (17) to update the critic. That is, $y_t$ is the RHS of (17) for RAA-Dueling-DQN.

**RAA-TD3.** For the case of TD3 where an actor and two critics are learned for deterministic policy and Q-value function respectively, the implementation of RAA is more complicated. Specifically, two critics $Q_{\theta_j}(j = 1, 2)$ are simultaneously trained with clipped double Q-learning. Then, the target values $y_{j,t}(j = 1, 2)$ for RAA-TD3 are given by

$$y_{j,t} = \beta \sum_{i=1}^{m} \widetilde{\alpha}_i \widehat{Q}_{\theta^i}(s_t, a_t) + (1 - \beta) \mathbb{E}_{s_{t+1}, r_t} \left[ r_t + \gamma \sum_{i=1}^{m} \widetilde{\alpha}_i \widehat{Q}_{\theta^i}(s_{t+1}, \pi_{\phi'}(s_{t+1}) + \epsilon) \right], \tag{19}$$

where $\widehat{Q}_{\theta^i}(s_t, a_t) = \min_{j=1,2} Q_{\theta_j^i}(s_t, a_t)$.

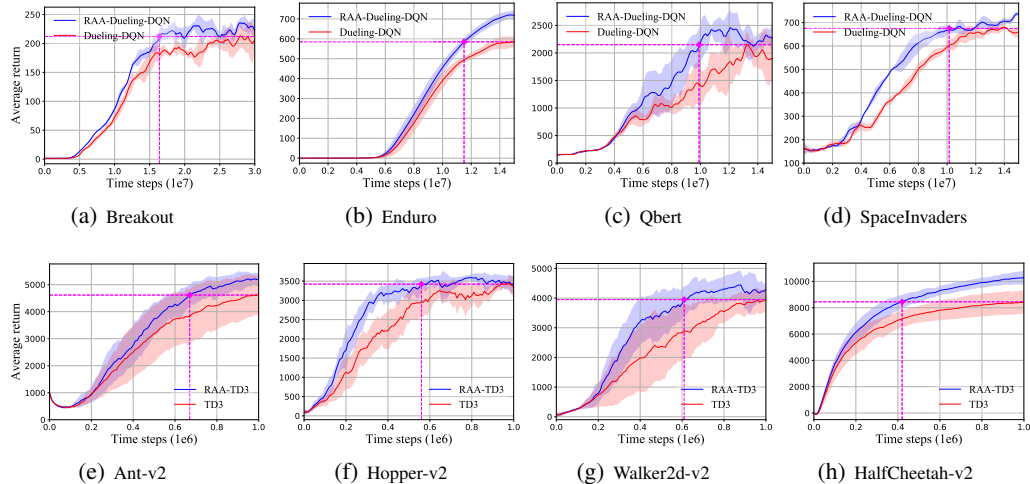

Figure 1: Learning Curves of Dueling-DQN, TD3 and their RAA variants on discrete and continuous control tasks. The solid curves correspond to the mean and the shaded region to the standard deviation over several trials. Curves are smoothed uniformly for visual clarity.

### 4.3.2  Adaptive restart

The idea of restarting an algorithm is well known in the numerical analysis literature. Vanilla Anderson acceleration has shown substantial improvements by incorporating with periodic restarts [30], where one periodically starts the acceleration scheme anew by only using information from the most recent iteration. In this section, to alleviate the problem that deep RL is notoriously prone to be trapped in local optimum, we propose an adaptive restart strategy for our RAA method.

Among the training steps of actor-critic with RAA, periodic restart checking steps are enforced to clear the memory immediately before the iteration completely crashes. More explicitly, the iteration is restarted whenever the average squared residual of current period exceeds the average squared residual of last period. Complete description of RAA-Dueling-DQN is summarized in Algorithm 1. And RAA-TD3 is given in Appendix B of the supplementary material.

## 5  Experiments

In this section, we present our experimental results and discuss their implications. We first give a detailed description of the environments (Atari 2600 and MuJoCo) used to evaluate our methods. Then, we report results on both discrete and continuous control tasks. Finally, we provide an ablative analysis for the proposed methodology. All default hyperparameters used in these experiments are listed in Appendix C of the supplementary material.

### 5.1  Experimental setup

**Atari 2600.**   For discrete control tasks, we perform experiments in the Arcade Learning Environment. We select four games (Breakout, Enduro, Qbert and SpaceInvaders) varying in their difficulty of convergence. The agent receives $84 \times 84 \times 4$ stacked grayscale images as inputs, as described in [1].

**MuJoCo.**   For continuous control tasks, we conduct experiments in environments built on the MuJoCo physics engine. We select a number of control tasks to evaluate the performance of the proposed methodology and the baseline methods. In each task, the agent takes a vector of physical states as input, and generates an action to manipulate the robots in the environment.

### 5.2  Comparative evaluation

To evaluate our RAA variant method, we select Dueling-DQN and TD3 as the baselines for discrete and continuous control tasks, respectively. Please note that we do not select DDPG as the baseline for continuous control tasks, as DDPG shows bad performance in difficult control tasks such as robotic manipulation. Figure 1 shows the total average return of evaluation rollouts during training for Dueling-DQN, TD3 and their RAA variants. We train five and seven different instances of each algorithm for Atari 2600 and MuJoCo, respectively. Besides, each baseline and corresponding RAA

variant are trained with same random seeds set and evaluated every 10000 environment steps, where each evaluation reports the average return over ten different rollouts.

The results in Figure 1 show that, overall, RAA variants outperform to corresponding baseline on most tasks with a large margin such as HalfCheetah-v2 and perform comparably to them on the easier tasks such as Enduro in terms of learning speed, which indicate that RAA is a feasible method to make existing off-policy RL algorithms more sample efficient. In addition to the direct benefit of acceleration mentioned above, we also observe that our RAA variants demonstrate superior or comparable final performance to the baseline methods in all tasks. In fact, RAA-Dueling-DQN can be seen as a weighted variant of Average-DQN [32], which can effectively reduce the variance of approximation error in the target values and thus shows improved performance. In summary, our approach brings an improvement in both the learning speed and final performance.

## 5.3 Ablation studies

The results in the previous section suggest that our RAA method can improve the sample efficiency of existing off-policy RL algorithms. In this section, we further examine how sensitive our approach is to the scaling of regularization. We also perform ablation studies to understand the contribution of each individual component: progressive update and adaptive restart. Additionally, we analyze the impact of different number of previous estimates $m$ and compare the behavior of our proposed RAA method over different learning rates.

**Regularization scale.** Our approach is sensitive to the scaling of regularization $\lambda$, because it control the norm of the coefficient vector and reduces the impact of approximation error. According to the conclusions of Proposition 1, larger regularization magnitude implies less impact of approximation error, but overlarge regularization will make the coefficients nearly identical and thus result in substantial degradation of acceleration performance. Figure 2 shows how learning performance changes on discrete control tasks when the regularization scale is varied, and consistent conclusion as above can be drawn from Figure 2. For continuous control tasks, it is difficult to obtain same conclusion due to the dominant effect of bias induced by sampling a minibatch relative to function approximation errors. Additional learning curves on continuous control tasks can be found in Appendix D of the supplementary material.

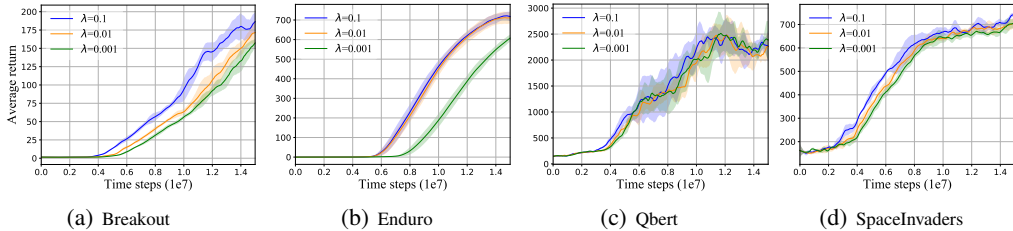

| (a) Breakout | (b) Enduro | (c) Qbert | (d) SpaceInvaders |

Figure 2: Sensitivity of RAA-Dueling-DQN to the scaling of regularization on discrete control tasks.

**Progressive update and adaptive restart.** This experiment compares our proposed approach with: (i) RAA without using progressive update (no progressive); (ii) RAA without adding adaptive restart (no restart); (iii) RAA without using progressive update and adding adaptive restart (no progressive and no restart). Figure 3 shows comparative learning curves on continuous control tasks. Although the significance of each component varies task to task, we see that using progressive update is essential for reducing the variance on all four tasks, consistent conclusion can also be drawn from Figure 1. Moreover, adding adaptive restart marginally improves the performance. Additional results on discrete control tasks can be found in Appendix D of the supplementary material.

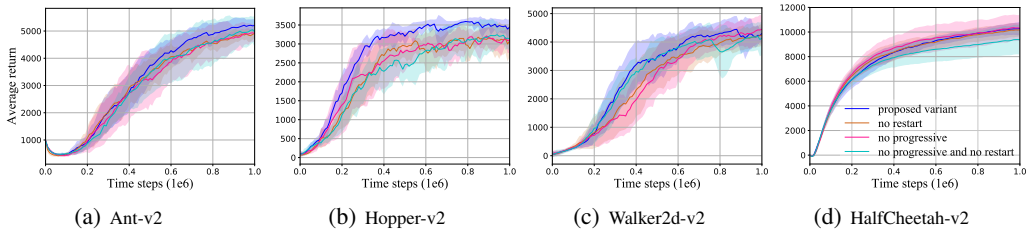

| (a) Ant-v2 | (b) Hopper-v2 | (c) Walker2d-v2 | (d) HalfCheetah-v2 |

Figure 3: Ablation analysis of RAA-TD3 (blue) over progressive update and adaptive restart.

**The number of previous estimates** $m$. In our experiments, the number of previous estimates $m$ is set to 5. In fact, there is a tradeoff between performance and computational cost. Fig.4 shows the results of RAA-TD3 using different $m(m = 1, 3, 5, 7, 9)$ on Walker2d task. Overall, we can conclude that larger $m$ leads to faster convergence and better final performance, but the improvement becomes small when $m$ exceeds a threshold. In practice, we suggest to take into account available computing resource and sample efficiency when applying our proposed RAA method to other works.

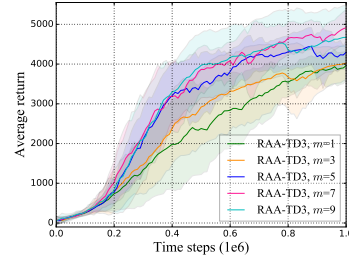

Figure 4: Learning Curves of RAA-TD3 on Walker2d-v2 with different $m$.

**Learning rate.** To compare the behavior of our proposed RAA method over different learning rates ($lr$), we perform additional experiments on Walker2d task, and the results of TD3 and our RAA-TD3 are shown in Fig.5. Overall, the improvement of our method is consistent across all learning rates, though the performance of both TD3 and our RAA-TD3 is bad under the setting with non-optimal learning rates, and the improvement is more significant when the learning rate is smaller. Moreover, consistent improvement of performance means that our proposed RAA method is effective and robust.

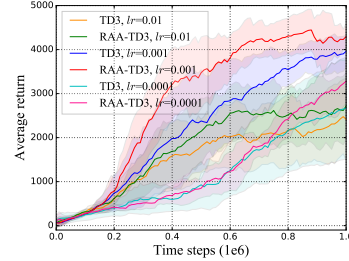

Figure 5: Performance comparison on Walker2d-v2 with different learning rates.

# 6 Conclusion

In this paper, we presented a general acceleration method for existing deep reinforcement learning (RL) algorithms. The main idea is drawn from regularized Anderson acceleration (RAA), which is an effective approach to speeding up the solving of fixed point problems with perturbations. Our theoretical results explain that vanilla Anderson acceleration can be directly applied to policy iteration under a tabular setting. Furthermore, RAA is extended to model-free deep RL by introducing an additional regularization term. Two rigorous bounds about coefficient vector demonstrate that the regularization term controls the norm of the coefficient vector produced by RAA and reduces the impact of perturbation induced by function approximation errors. Moreover, we verified that the proposed method can significantly accelerate off-policy deep RL algorithms such as Dueling-DQN and TD3. The ablation studies show that progressive update and adaptive restart strategies can enhance the performance. For future work, how to combine Anderson acceleration or its variants with on-policy deep RL is an exciting avenue.

# Acknowledgments

Gao Huang is supported in part by Beijing Academy of Artificial Intelligence (BAAI) under grant BAAI2019QN0106 and Tencent AI Lab Rhino-Bird Focused Research Program under grant JR201914. This research is supported by the National Science Foundation of China (NSFC) under grant 41427806.

## Footnotes

[2]Notice that we don't impose a positivity condition on the coefficients.

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
