[Supplementary Material]

# Supplementary Material

## A  Proofs

### A.1  Solution to Anderson Acceleration

*Proof.* Let $\mu$ be the dual variable of the equality constraint of (10). Both $\alpha^k$ and $\mu^k$ should satisfy the Karush-Kuhn-Tucker (KKT) system

$$\begin{bmatrix} 2\Delta_k^T\Delta_k & \mathbf{1} \\ \mathbf{1}^T & 0 \end{bmatrix}\begin{bmatrix} \alpha^k \\ \mu^k \end{bmatrix} = \begin{bmatrix} \mathbf{0} \\ 1 \end{bmatrix}. \tag{20}$$

This block matrix can be inverted explicitly, with

$$\begin{bmatrix} 2\Delta_k^T\Delta_k & \mathbf{1} \\ \mathbf{1}^T & 0 \end{bmatrix}^{-1} = \frac{1}{\mathbf{1}^T(\Delta_k^T\Delta_k)^{-1}\mathbf{1}}\begin{bmatrix} \frac{1}{2}(\Delta_k^T\Delta_k)^{-1}Y_k & (\Delta_k^T\Delta_k)^{-1}\mathbf{1} \\ \mathbf{1}^T(\Delta_k^T\Delta_k)^{-1} & -2 \end{bmatrix}, \tag{21}$$

where $Y_k = \mathbf{1}^T(\Delta_k^T\Delta_k)^{-1}\mathbf{1}I - \mathbf{1}I^T(\Delta_k^T\Delta_k)^{-1}$. Using this inverse we easily solve the linear system, which gives the result in (7).  □

### A.2  Proof to Proposition 1

*Proof.* We begin by the bound on $\widetilde{\alpha}^k$. Indeed, with (14),

$$\|\widetilde{\alpha}^k\|^2 = \frac{\mathbf{1}^T(\widetilde{\Delta}_k^T\widetilde{\Delta}_k + \lambda I)^{-2}\mathbf{1}}{(\mathbf{1}^T(\widetilde{\Delta}_k^T\widetilde{\Delta}_k + \lambda I)^{-1}\mathbf{1})^2} \tag{22}$$

$$\leq \frac{1}{m}\max_{\|v\|=1}\frac{v^T(\widetilde{\Delta}_k^T\widetilde{\Delta}_k + \lambda I)^{-2}v}{(v^T(\widetilde{\Delta}_k^T\widetilde{\Delta}_k + \lambda I)^{-1}v)^2} \tag{23}$$

$$= \frac{1}{m}\max_{\|v\|=1}\frac{\|(\widetilde{\Delta}_k^T\widetilde{\Delta}_k + \lambda I)^{-\frac{1}{2}}(\widetilde{\Delta}_k^T\widetilde{\Delta}_k + \lambda I)^{-\frac{1}{2}}v\|^2}{\|(\widetilde{\Delta}_k^T\widetilde{\Delta}_k + \lambda I)^{-\frac{1}{2}}v\|^4} \tag{24}$$

$$\leq \frac{1}{m}\|(\widetilde{\Delta}_k^T\widetilde{\Delta}_k + \lambda I)^{-\frac{1}{2}}\|^2\max_{\|v\|=1}\frac{1}{\|(\widetilde{\Delta}_k^T\widetilde{\Delta}_k + \lambda I)^{-\frac{1}{2}}v\|^2} \tag{25}$$

$$= \frac{1}{m}\|(\widetilde{\Delta}_k^T\widetilde{\Delta}_k + \lambda I)^{-\frac{1}{2}}\|^2\|(\widetilde{\Delta}_k^T\widetilde{\Delta}_k + \lambda I)^{\frac{1}{2}}\|^2 \tag{26}$$

$$\leq \frac{\lambda + \|\widetilde{\Delta}_k\|^2}{m\lambda}, \tag{27}$$

where the last inequality is because $\widetilde{\Delta}_k^T\widetilde{\Delta}_k \geq 0$, we have $(\widetilde{\Delta}_k^T\widetilde{\Delta}_k + \lambda I) \geq \lambda I$.

We will bound $\widetilde{\alpha}^k - \alpha^k$ from now on. Let $\widetilde{\mu}^k$ be the dual variable of the equality constraint in (13), then $\widetilde{\alpha}^k$ and $\widetilde{\mu}^k$ should satisfy the KKT system

$$\begin{bmatrix} 2(\widetilde{\Delta}_k^T\widetilde{\Delta}_k + \lambda I) & \mathbf{1} \\ \mathbf{1}^T & 0 \end{bmatrix}\begin{bmatrix} \widetilde{\alpha}^k \\ \widetilde{\mu}^k \end{bmatrix} = \begin{bmatrix} \mathbf{0} \\ 1 \end{bmatrix}. \tag{28}$$

Expanding the LHS of (28), we obtain

$$\begin{bmatrix} 2(\widetilde{\Delta}_k^T\widetilde{\Delta}_k + \lambda I) & \mathbf{1} \\ \mathbf{1}^T & 0 \end{bmatrix}\begin{bmatrix} \widetilde{\alpha}^k \\ \widetilde{\mu}^k \end{bmatrix} = \begin{bmatrix} 2\Delta_k^T\Delta_k & \mathbf{1} \\ \mathbf{1}^T & 0 \end{bmatrix}\begin{bmatrix} \alpha^k \\ \mu^k \end{bmatrix} + \begin{bmatrix} 2\Delta_k^T\Delta_k & \mathbf{1} \\ \mathbf{1}^T & 0 \end{bmatrix}\begin{bmatrix} \widetilde{\alpha}^k - \alpha^k \\ \widetilde{\mu}^k - \mu^k \end{bmatrix}$$
$$+ \begin{bmatrix} 2(\widetilde{\Delta}_k^T\widetilde{\Delta}_k + \lambda I - \Delta_k^T\Delta_k) & \mathbf{0} \\ \mathbf{0}^T & 0 \end{bmatrix}\begin{bmatrix} \widetilde{\alpha}^k \\ \widetilde{\mu}^k \end{bmatrix}. \tag{29}$$

Using the condition (28) and (20), the system becomes

$$\begin{bmatrix} 2(\widetilde{\Delta}_k^T\widetilde{\Delta}_k + \lambda I) & \mathbf{1} \\ \mathbf{1}^T & 0 \end{bmatrix}\begin{bmatrix} \widetilde{\alpha}^k - \alpha^k \\ \widetilde{\mu}^k - \mu^k \end{bmatrix} = -\begin{bmatrix} 2(\widetilde{\Delta}_k^T\widetilde{\Delta}_k + \lambda I - \Delta_k^T\Delta_k)\alpha^k \\ 0 \end{bmatrix}. \tag{30}$$

The explicit solution is obtained by inverting the block matrix, and is written

$$\widetilde{\alpha}^k - \alpha^k = -\left(I - \frac{(\widetilde{\Delta}_k^T\widetilde{\Delta}_k + \lambda I)^{-1}\mathbf{1}\mathbf{1}^T}{\mathbf{1}^T(\widetilde{\Delta}_k^T\widetilde{\Delta}_k + \lambda I)^{-1}\mathbf{1}}\right)(\widetilde{\Delta}_k^T\widetilde{\Delta}_k + \lambda I)^{-1}(\widetilde{\Delta}_k^T\widetilde{\Delta}_k + \lambda I - \Delta_k^T\Delta_k)\alpha^k. \tag{31}$$

Then, we can bound the norm of $\widetilde{\alpha}^k - \alpha^k$ by

$$\|\widetilde{\alpha}^k - \alpha^k\| \leq \left\| I - \frac{(\widetilde{\Delta}_k^T \widetilde{\Delta}_k + \lambda I)^{-1} \mathbf{1}\mathbf{1}^T}{\mathbf{1}^T (\widetilde{\Delta}_k^T \widetilde{\Delta}_k + \lambda I)^{-1} \mathbf{1}} \right\| \left\| (\widetilde{\Delta}_k^T \widetilde{\Delta}_k + \lambda I)^{-1} \right\| \left\| (\widetilde{\Delta}_k^T \widetilde{\Delta}_k + \lambda I - \Delta_k^T \Delta_k) \right\| \left\| \alpha^k \right\| \quad (32)$$

$$\leq \left\| (\widetilde{\Delta}_k^T \widetilde{\Delta}_k + \lambda I)^{-1} \right\| \left\| (\widetilde{\Delta}_k^T \widetilde{\Delta}_k + \lambda I - \Delta_k^T \Delta_k) \right\| \left\| \alpha^k \right\| \quad (33)$$

$$\leq \frac{\|\widetilde{\Delta}_k^T \widetilde{\Delta}_k - \Delta_k^T \Delta_k\| + \lambda}{\lambda} \|\alpha^k\|, \quad (34)$$

which is the desired result. $\qquad\square$

## B  RAA-TD3

---
**Algorithm 2:** RAA-TD3 Algorithm
---
Initialize critic networks $Q_{\theta_1}, Q_{\theta_2}$, and actor network $\pi_\phi$ with random parameters $\theta_1, \theta_2, \phi$;
Initialize target networks $\theta_j^i \leftarrow \theta_j$ $(i = 1, ..., m; j = 1, 2)$, $\phi' \leftarrow \phi$;
Initialize replay buffer $\mathcal{D}$;
Initialize restart checking period $T_r$ and maximum training steps $K$;
Set $k = 0$, $c_1 = 1$, $\Delta_{min} = \inf$, $\Delta_{T_r} = 0$;
**while** $k < K$ **do**
    Receive initial observation state $s_0$;
    **for** $t = 1$ *to* $T$ **do**
        Set $k = k + 1$, and $m_k = \min(c_k, m)$;
        Select action $a_t$ with exploration noise $a \sim \pi_\phi(s) + \epsilon$, $\epsilon \sim \mathcal{N}(0, \sigma)$;
        Execute $a_t$, receive $r_t$ and $s_{t+1}$, store transition $(s_t, a_t, r_t, s_{t+1})$ into $\mathcal{D}$;
        Sample minibatch of $N$ transitions $(s, a, r, s')$ from $\mathcal{D}$;
        Perform Anderson acceleration steps (13)-(14)and obtain $\widetilde{\alpha}^k$, $\Delta_{T_r} = \Delta_{T_r} + \|\widetilde{\delta}_k\|_2^2$;
        Update critic networks by minimizing the loss function (18) with (19);
        **if** $t \mod M = 0$ **then**
            Update actor network by the deterministic policy gradient:
                $\nabla_\phi J(\phi) = N^{-1} \sum \nabla_a Q_{\theta_1}(s, a)|_{a = \pi_\phi(s)} \nabla_\phi \pi_\phi(s)$;
            Update target networks:
                $\theta_j^i \leftarrow \theta_j^{i+1}$, $\theta_j^m \leftarrow \tau \theta_j + (1 - \tau)\theta_j^m$ $(i = 1, ..., m - 1; j = 1, 2)$ ;
                $\phi' \leftarrow \tau \phi + (1 - \tau)\phi'$;
        $c_{k+1} = c_k + 1$;
        **if** $k \mod T_r = 0$ **then**
            $\Delta_{min} = \min(\Delta_{min}, \Delta_{T_r})$;
            **if** $\Delta_{T_r} > \Delta_{min}$ **then**
                $\Delta_{min} = \inf$, and $c_{k+1} = 1$;
---

# C  Hyperparameters

Table 1: Hyperparameters used in Dueling-DQN and RAA-Dueling-DQN.

| Hyperparameters | Value |
|---|---|
| *Network* | |
|     channels | 32, 64, 64 |
|     filter size | $8 \times 8, 4 \times 4, 3 \times 3$ |
|     stride | 4, 2, 1 |
|     Val: (hidden units, output units) | (512, 1) |
|     Adv: (hidden units, output units) | (512, action dimensions) |
| *Shared* | |
|     optimizer | RMSprop |
|     start time steps | $5 \times 10^4$ |
|     discount factor | 0.99 |
|     replay buffer size | $10^6$ |
|     batch size | 32 |
|     frames stacked | 4 |
|     action repetitions | 4 |
|     learning rate | 0.00025 |
| *RAA-Dueling-DQN* | |
|     progressive coefficient ($\beta$) | 0.05 |
|     sample size for RAA ($N_A$) | 128 |
|     regularization scale | 0.1 |
|     number of previous estimates | 5 |
|     target update interval | 2000 |
| *Dueling-DQN* | |
|     target update interval | 10000 |

Table 2: Hyperparameters used in TD3 and RAA-TD3.

| Hyperparameters | Value |
|---|---|
| *Network* | |
|     Critic: hidden units | 400, 300 |
|            output units | 1 |
|     Actor: hidden units | 400, 300 |
|            output units | action dimensions |
| *Shared* | |
|     optimizer | Adam |
|     start time steps | $10^4$ |
|     discount factor | 0.99 |
|     replay buffer size | $10^6$ |
|     batch size | 100 |
|     exploration noise | 0.1 |
|     target update rate ($\tau$) | $5 \times 10^{-3}$ |
|     actor update frequency | 2 |
|     exploration policy | $\mathcal{N}(0, 0.2)$ |
| *RAA-TD3* | |
|     progressive coefficient ($\beta$) | 0.1 |
|     sample size for RAA ($N_A$) | 400 |
|     regularization scale | 0.001 |
|     number of previous estimates | 5 |
| *TD3* | |

# D  Additional Learning Curves

(a) Ant-v2    (b) Hopper-v2    (c) Walker2d-v2    (d) HalfCheetah-v2

Figure 6: Sensitivity of RAA-TD3 to the scaling of regularization on continuous control tasks.

(a) Breakout    (b) Enduro    (c) Qbert    (d) SpaceInvaders

Figure 7: Ablation analysis of RAA-Dueling-DQN (blue) over progressive update and adaptive restart.