[Reviews · NeurIPS 2019]

Reviewer 1



This paper is very inspiring. Slow convergence and sample inefficiency remain challenging problems in RL, especially when handling continuous and high-dimensional state spaces. This paper applies a general method that drawn from regularized Anderson acceleration (RAA) to accelerate the convergence or improve the sample efficiency for the model-free, off-policy deep RL. Overall this paper is well constructed and improves the problem from a novel point of view (they make the observation that RL is closely linked to fixed-point iteration). The results show that their approach substantially improves both the learning speed and final performance of state-of-the-art deep RL algorithms, and the literature review shows that the author is knowledgeable in this field. Here are my major concerns: I would suggest adding a description about policy iteration is missing in 3.2 since value iteration is a special variant of policy iteration. Some minor suggestions: Line 116 to 125: specify m > 1. Line 149: Besides l2-norm, do other methods (like l1-norm or l∞-norm) works effectively here? Typo on Appendix ??: line 151, line 191, line 239, line 245, line 286, line 294.

Reviewer 2



The main contribution of this paper is to apply Anderson acceleration to the setting of deep reinforcement learning. The authors first propose a regularized form of Anderson acceleration, and then show how it can be applied to two practical deep RL algorithms: DQN and TD3. Originality: This paper falls under the vein of applying existing techniques to a novel domain. While the idea of introducing Anderson acceleration to the context of RL is not new, as the authors mention, it has not been applied to deep RL methods. While the originality is somewhat limited in this aspect, developing a practical and functional improvement for deep RL algorithms is not trivial. Quality: The paper is technically sound, and the experimental analysis is fair and supports the main thesis of the paper. I like the fact that introducing RAA results in similar performance gains on both TD3 and Dueling-DQN. This is promising from a reproducibility standpoint, as it is more likely that the performance gain from RAA is real and meaningful, and not simply resolving an odd quirk found in a particular deep RL method. Clarity: The paper is well-written. The background on Anderson acceleration is clearly explained, as well as the motivation and extension to the deep RL domain. Significance: Developing good and stable off-policy RL algorithms is an important area of research. This work proposes a well-motivated modification to existing off-policy algorithms which appears to be simple to implement and has the promise of moderate performance gains. I believe this work will be of interest to the deep RL community. Minor typos and clarifications: - Appendix links are broken (i.e. Lines 191, 239, 286, 294, and more) - It is not immediately clear what value of "m" was used for the experiments. I assume "m" corresponds to the "number of previous estimates" hyperparameter, but this could be made more explicit. Additionally, there is some informal language used in the paper. I have proposed some minor modifications: Line 35: "this mapping is essentially a kind of fixed-point problem" -> "Iterating this mapping results in a fixed-point problem" Line 173: "may make the iteration get stuck" -> delete?

Reviewer 3



Clarity: The writing in the paper is clear. The presentation of Anderson acceleration and the proposed bound on the iterates is also clear. Originality: The contributions in this paper appear novel and past work is appropriately cited. Significance & Quality: The performance benefits of this method are not sufficiently clear from the chosen experiments. As presented in this paper, the key idea in Anderson acceleration is to use the iterates in a contraction to move faster towards the fixed point. The performance benefit over the baselines in Figure 1 appear small. These benefits could be conflated with other factors such as the choice of step-size, and the optimizer used. The experiments do not disentangle these conflating factors. In particular, common deep-learning optimizers (ADAM, RMSProp) contain momentum terms that could (weakly) mimic some of the benefits of Anderson acceleration in combining information across iterates for faster convergence. There should be experiments that examine this.

[Author Response · NeurIPS 2019]

First of all, we would like to thank all reviewers for their insightful comments and suggestions!

## Reviewer #1

**Adding a description about policy iteration in Sec 3.2.** Thanks for the suggestion. We will add a paragraph to
provide background materials on policy iteration.

**Do other norms work?** Yes, we can also use $L_1$-norm or $L_\infty$-norm, in which case the optimization problem becomes
a linear program. But in the field of reinforcement learning (RL), $L_2$-norm is the most common choice due to its
efficiency and effectiveness. Thus we adopt $L_2$-norm in the paper to ensure consistency between the objective of
Anderson acceleration (AA) and the loss of Q-value function (critic).

**Minors.** Thanks! We will fix these issues in the revision.

## Reviewer #2

**Impact of the number of previous estimates $m$.** Indeed, there is a tradeoff
between performance and computational cost. We have analyzed the impact of
using different $m$ during the rebuttal period, and part of the results are shown
in Fig.1. Overall, larger $m$ leads to better performance, but the improvement
becomes small when $m$ exceeds a threshold. These additional experiments
will be added to Sec. 5.3 (ablation studies) in the revision.

**Value of $m$.** We set $m$ to 5 in our experiments. The detailed hyperparameter
settings are given in Appendix C, where "number of previous estimates"
corresponds to $m$. (Sorry for the broken link in Line 245.)

Figure 1: Performance of RAA-TD3 on Walker2d-v2 with different $m$.

**Impact of the error/perturbations on the final solution found by RAA?**
Indeed, it is meaningful to construct error bounds for the value function.
However, this seems to be difficult in the context of *deep* RL. First, the approximation error of value function largely
originates from the deep neural networks, for which the generalization error bound is difficult to obtain. Second, there
is no explicit connection between the error of coefficient vector $\alpha$ and the error of value function. Fortunately, for the
coefficient vector $\alpha$, we can still provide a clear connection between regularization and the approximation error, as
shown by Proposition 1 in line 187-197. Constructing error bounds for value function under the setting of linear or
other interpretable function approximators is an interesting topic for future work.

**Minors.** Thanks for your detailed comments! We will carefully fix all the minor issues into our revision.

## Reviewer #3

**Performance benefit.** Please note that our motivation of introducing RAA is
to improve the *sample efficiency* (convergence speed) of deep RL, instead of
improving the final performance. Fig.1 in our paper shows that RAA-based
RL algorithms generally require half the number of samples to achieve com-
parable performance as the counterparts without RAA, which is a significant
boost in terms of *efficiency*. Interestingly, due to variance reduction of ap-
proximation error in the target values, our algorithm also improves the final
performance in most of the cases. In other words, RAA not only substantially
improves the sample efficiency of deep RL, but also boosts the final perfor-
mance in many cases.

Figure 2: Performance comparison on Walker2d-v2 with different learning rates.

**Conflating factors.** In fact, we have tried our best to isolated all conflating
factors in our experiments: we picked state-of-the-art (SOTA) deep RL al-
gorithm and simply add the proposed RAA module to it *without changing any of the hyperparameters* (including
step-size and the optimizer). This means that (1) our baselines are very competitive; (2) we used exactly the same
hyperparameters as the baselines; and (3) the only difference between our algorithm and the baselines is using or not
using RAA. Therefore, we believe our experiments are fair.

**A sweep of step-sizes.** During the rebuttal period, we performed additional experiments to compare the behavior of
RAA over different learning rates ($lr$), and the results are shown in Fig.2. Overall, the improvement of our method is
consistent across all learning rates, and the improvement is more significant when the learning rate is smaller. Additional
experiments will be added to Sec. 5.3 (ablation studies) in the revision.

**Momentum terms could mimic some benefits of RAA?** Indeed, momentum also aggregates information across
iterations and leads to faster convergence. But as noted above, our baselines are SOTA deep RL algorithms, which
are already equipped with advanced momentum-based optimizer (e.g., ADAM). This means RAA is compatible with
momentum and can further speed up the convergence.

**Assess RAA in deep RL regime.** This seems to be a misunderstanding. We actually focus on deep RL in this work.

[Meta-Review · NeurIPS 2019]

This work is an interesting contribution to deep RL that considers using Anderson acceleration to improve off-policy TD based algorithms. The approach is supported by some theory as well as experiments on standard benchmark problems. Overall, reviewers like the paper and agree it should be accepted.